# Feasibility Study on the Discrimination of Fluor Concentration in the Liquid Scintillator Using PMT Waveform and Short-Pass Filter

**DOI:** 10.3390/s23052728

**Published:** 2023-03-02

**Authors:** Na-Ri Kim, Kyung-Kwang Joo, Hyun-Gi Lee

**Affiliations:** Center for Precision Neutrino Research, Department of Physics, Chonnam National University, Yongbong-ro 77, Puk-gu, Gwangju 61186, Republic of Korea

**Keywords:** liquid scintillator, linear alkyl benzene, fluor, PMT, waveform, short-pass filter

## Abstract

Neutrinos are difficult to detect because they weakly interact with matter, making their properties least known. The response of the neutrino detector depends on the optical properties of the liquid scintillator (LS). Monitoring any characteristic changes in the LS helps to understand the temporal variation of detector response. In this study, a detector filled with LS was used to study the characteristics of the neutrinos detector. We investigated a method to distinguish the concentrations of PPO and bis-MSB, which are fluors added to LS, through a photomultiplier tube (PMT) acting as an optical sensor. Conventionally, it is very challenging to discriminate the flour concentration dissolved in LS. We employed the information of pulse shape and PMT coupled with the short-pass filter. To date, no literature report on a measurement using such an experimental setup has been published. As the concentration of PPO was increased, changes in the pulse shape were observed. In addition, as the concentration of bis-MSB was increased, a decrease in the light yield was observed in the PMT equipped with the short-pass filter. This result suggests the feasibility of real-time monitoring of LS properties, which are correlated with the fluor concentration, using a PMT without extracting the LS samples from the detector during the data acquisition process.

## 1. Introduction

Among the elementary particles of the Standard Model, neutrinos are one of the least understood particles. Because of their weak interaction with matter, accumulating sufficient statistics of neutrino events is a time-intensive and challenging task. To circumvent these limitations, the neutrino detector is typically exposed to the source until sufficient statistics are obtained [1,2]. In the last decade, several neutrino detectors have been used in various pioneering experiments, and various next-generation massive detectors have been constructed to operate for several years for gaining deeper insights into the nature of neutrinos [1,2,3].

The response of the detector is directly coupled with the light yield in the LS used for the neutrino target [4]. According to previous studies, during several years of operation of the detector, degradation of light yield in LS was reported [5]. Therefore, confirming the stability of a substance used for a target and monitoring its properties are important for better understanding and operating stable detectors. Since extracting LS from the detector during its operation is difficult, developing an indirect method to monitor the properties of the LS in real time will help to understand the detector response in operation.

The light yield is one of the primary properties of the LS and is regularly monitored for checking detector performance in the middle of detector operation [6]. Distinguishing the concentration of dissolved fluor in LS through the light yield at a reasonable level is very challenging [7,8]. However, it is known that the emission spectrum and the decay time of LS are related to the concentration of 2,5-diphenyloxazole (C_15_H_11_NO, PPO) and 1,4-bis(2-methylstyryl)benzene (C_24_H_22_, bis-MSB) dissolved in the LS [9,10]. In this study, we present a method to distinguish the concentration of the fluor dissolved in the LS in real time using the signal obtained by the photomultiplier tube (PMT) used as the sensor of the detector. We utilized the waveform information of the PMT affected by the decay time of the LS and adopted a short-pass filter to the PMT to discriminate the change in the emission spectrum of the LS.

In Section 2, LS sample preparation, appearance with UV light, and emission spectrum with different PPO and bis-MSB concentration are described. An experimental setup for discrimination of fluor concentration in the LS was shown in Section 3. A light yield, waveform, and short-passed ratio of LS were measured, and the obtained characteristics were presented with their concentration in Section 4. Finally, a summary and outline of future prospects are written in Section 5.

## 2. Preparation of LS Samples

Linear alkyl benzene (LAB, C_n_H_2n+1_-C_6_H_5_, n = 10–13) provides a relatively high light yield and long attenuation distance due to the high transparency [11]. It has a high flash point and is environmentally friendly [12]. For those reasons, LAB has been used as a scintillator solvent in many experiments over the last decades [11,12,13,14]. The LAB-based LS is synthesized by mixing wavelength-shifting fluor with an aromatic solvent of LAB. In many neutrino experiments, the LS was synthesized by mixing LAB with PPO and bis-MSB, which act as wavelength shifters. A wavelength of ~280 nm photon emitted from the LAB is converted to a wavelength around ~380 nm by the primary wavelength shifter PPO and the emitted lights are converted again to ~420 nm [15], by the secondary wavelength shifter bis-MSB, where near the maximum quantum efficiency of the PMT [7,10]. The light yield of LS is an essential property for the detectors, especially for the reconstruction of the particle [16]. A higher light yield provides more information on the event. The light yields, decay time, and emission spectrum of LS depend on the concentration of the fluors dissolved in the LS [9]. The fluor concentration is different for the experiments; for example, PPO of 2–7 g/L and bis-MSB of 1–30 mg/L is used for the concentration [6,11,12,13].

Table 1 summarizes various LS samples with different PPO concentrations. The PPO concentration ranges from 0.5 to 10 g/L. Table 2 summarizes the LS with different bis-MSB concentrations from 1 to 50 mg/L. In ref. [12], the LS has the highest light yield at the PPO concentration of 3 g/L. Accordingly, the PPO concentrations of the sample in Table 2 were optimized and adjusted to 3 g/L. Further details for the light yield measurement of the samples in Table 1 and Table 2 can be found in Section 4.1.

For a quick eye inspection, the samples were illuminated by an Ultraviolet (UV) lamp at 312 nm. Figure 1 shows the emission of scintillation lights of LS samples from the excitation of UV light with the digital camera (EOS D series, Manufacturer: Canon, Tokyo, Japan). It is almost impossible to identify the wavelength difference with the naked eye.

The emission spectra of the LS samples with the lowest (A1, B1) and the highest (A8, B8) PPO and bis-MSB concentrations were measured using a fluorescence spectrophotometer (Cary Eclipse fluorescence spectrometer, Manufacturer: Varian Australasia, Sydney, Australia) and cross-checked by another institute (KONKUK university TECH research facilities: Seoul, Republic of Korea). As described earlier, the primary fluor PPO and secondary fluor bis-MSB have maximum emissions at ~380 nm, and ~420 nm, respectively [12]. In Figure 2a, the hatched area represents the energy transition region where absorption and emission occur between PPO and bis-MSB. Because the x-axis is stretched, the emission spectra obtained at two extreme PPO or bis-MSB concentrations between (A1 and A8) or (B1 and B8) may be distinguished, although the process is challenging. Compared to the image seen by the naked eye in Figure 1, a slight shift in the emission spectrum is evident with the increasing fluor concentration. The emission spectrum shifts to the right for the sample with a high PPO concentration. Similarly, the emission spectra of the samples with a high bis-MSB concentration shift to the right.

As the PPO or bis-MSB concentration increases, it can be seen that the left side of the distribution shifts relative to the right and rapidly rises. This suggests the possibility of determining the concentration of PPO or bis-MSB according to the degree of wavelength shift. To demonstrate this, the response of the PMT with and without using a 425 nm short-pass filter (High-performance optical density 4.0 short-pass filters, Manufacturer: Edmund Optics, Barrington, IL, USA) was investigated and compared.

## 3. Experimental Setup

The energy of the neutrinos is converted into scintillation photons by ionization in the LS and detected by the PMTs installed in the detectors [18]. PMTs are optical sensors that are widely used for detecting particles, and several PMTs are employed in a single detector [19]. These PMTs have a fast response in the nano-second regime and have high quantum efficiency around the 300–500 nm region [20]. The changes in the emission spectra due to the different concentrations of bis-MSB and PPO were observed by using a 425 nm short-pass filter in the PMT as shown in Figure 3.

Figure 3 shows that the scintillation light emitted from the vial containing the LS sample is observed through a polytetrafluoroethylene (PTFE) coupler and two 2-inch PMTs. The scintillation light of the LS sample is produced from the Compton scattering of the gamma emitted by the ^60^Co source. A Teflon cylindrical tube was placed in the middle to prepare for contact. Teflon cylindrical tube made of PTFE has a length of 7.5 cm and an inner diameter of 2 inches. The gain of two 2-inch in diameter PMTs (H7195, Manufacturer: Hamamatsu, Shizuoka, Japan), was adjusted to a typical value of 3.0 × 10^6^, and the waveform emitted from the two PMTs was digitized by a Flash-Analog-to-Digital-Converter (FADC 400 MHz, Manufacturer: Notice, Seoul, Republic of Korea) of 10-bit resolution. The quantum efficiency of the H7195 PMT is shown in Figure 2b [17]. To measure the change in emission spectrum depending on fluor concentration, a 425 nm short-pass filter [21] with the transmittance of Figure 2c was attached to the 2-inch PMT-B in Figure 3b. The short-pass filter is made with fused silica and has a diameter of $50.0_{-0.2}^{+0.0}$ mm.

## 4. Measurements & Results

The light yield, pulse shape, and wavelength shift of the LS sample with different fluor concentrations were measured using the following method.

### 4.1. Light Yield Measurement

The light yield of an LS varies with the concentration of PPO and bis-MSB present in the LS [12]. The light yield for each sample in Table 1 and Table 2 was measured by the fitting of the Compton edge produced from the backscattering of the gamma emitted by ^60^Co. Figure 4 shows the fitting of the photoelectrons (PE) distribution to determine the Compton edge. The light yield of the scintillator sample is determined from the summation of the number of PE measured by PMT-A and PMT-B in Figure 3a.

The scintillation light yield of each sample was observed. Figure 5a presents the obtained light yield of the samples shown in Table 1. The light yield is maximum at a PPO concentration of 3 g/L and slowly decreases at higher concentrations. Figure 5b shows the observed light yield of the samples listed in Table 2. These samples have different bis-MSB concentrations, but the same PPO concentration (3 g/L). The scintillation light yield saturates at a bis-MSB concentration of ~10 mg/L.

### 4.2. Pulse Shape Measurement

It is known that the decay time of LS varies depending on the PPO concentration added as a fluor. The waveform of the H7195 PMT, consisting of three components (a rising flank, peak, and tail), responds sensitively to the timing of the scintillation light incident on the photocathode [19,22]. The rising flank and peak sensitively respond to the scintillator’s fast decay component, while the tail is sensitive to the slow decay component [22,23]. Figure 6 shows the area normalized pulse shape with the different fluor concentrations. The difference in the PPO concentration discriminates the observed waveforms; however, only slight changes in the waveform are observed with the changing bis-MSB concentration. This suggests the possibility of distinguishing the concentration of PPO by using waveform information.

### 4.3. Wavelength Shift Measurement

The emission spectrum analyzed in Section 2 reveals that the emitted wavelength changes with the PPO and bis-MSB concentration. As previously demonstrated in Figure 3b, a short-pass filter was inserted only into PMT-B to distinguish the concentration of the fluor based on the changes in the emission spectrum. Figure 7 compares the observed PE ratio between PMT-A and PMT-B with and without the short-pass filter. Without the filter, the differences in the distribution are indistinct irrespective of the fluor concentration. However, when the filter is applied to the PMT, the distribution is distinguished from the changes in the concentration. The observed PE ratio between the two H7195 PMT discriminated as fluor concentration increased.

### 4.4. Observed LS Response with Different Fluor Concentration

The pulse shape and wavelength shift in the emission spectrum are discussed in the previous section with different PPO and bis-MSB concentrations. The measured responses for each sample are summarized in Figure 8.

In Section 4.2, the area-normalized pulse shape at different fluor concentrations of each sample is discussed. Figure 8a shows the pulse height/pulse area plotted against the fluor concentration. In the case of PPO, the pulse height/pulse area increases until the concentration reaches ~4 g/L, and then becomes saturated. Figure 8b shows the pulse height/pulse area according to bis-MSB concentration. The area-normalized pulse height changes only slightly with the changing concentration. Note that the amount of bis-MSB is very small compared to that of PPO and the changes caused by the bis-MSB concentration are negligible.

In Section 4.3, the observed PE ratio passing through the short-pass filter was measured using samples with different PPO and bis-MSB concentrations. Figure 8c,d shows the observed PE ratio with the different PPO and bis-MSB concentrations. In Figure 8c, the amount of change in the y-axis as the PPO concentration increases is not evident. However, a clear decrease in the percentage of scintillation lights passing through the short-pass filter was observed with the bis-MSB concentration, as shown in Figure 8d. These results indicate the feasibility of distinguishing bis-MSB concentration, which was quite challenging with a conventional method based on the light yield measurement.

## 5. Summary

Neutrino energy is typically reconstructed by the number of photons observed by the PMTs mounted in the detector. The number of observed photons is directly coupled with the scintillation light yield of the LS. Comprehending the characteristics of the LS provides helpful information for a better understanding of the detector response. The earlier measurements were focused on the characteristics of the scintillation light yield of the LS depending on the fluor concentration. However, both PPO and bis-MSB are correlated with the light yield, and the two fluor concentrations cannot be uniquely determined from the light yield characteristic of LS. In this study, the additional featured characteristics that are sensitive to the fluor concentration, the waveform, and changes in the emission spectrum have been selected and measured. We observed that the area normalized pulse height changes depending on the PPO concentration. In the bis-MSB case, a decrease in the ratio between the PE of two PMTs was observed using a short-pass filter.

With different PPO concentrations, the changes in the entire pulse shape were observed in addition to the changes in the pulse height. When entire pulse shapes, emission spectrum changes, and the light yield are employed instead of using only the pulse height, more information from the higher dimension is expected to improve the discrimination power. The additional features related to the concentration can be extracted using machine learning techniques, such as an artificial neural network [23]. We anticipate that our method can be used to monitor the characteristics of LS in the real-time operation of PMT-based detectors in the future [1,24,25].

## Figures and Tables

**Figure 1 sensors-23-02728-f001:**
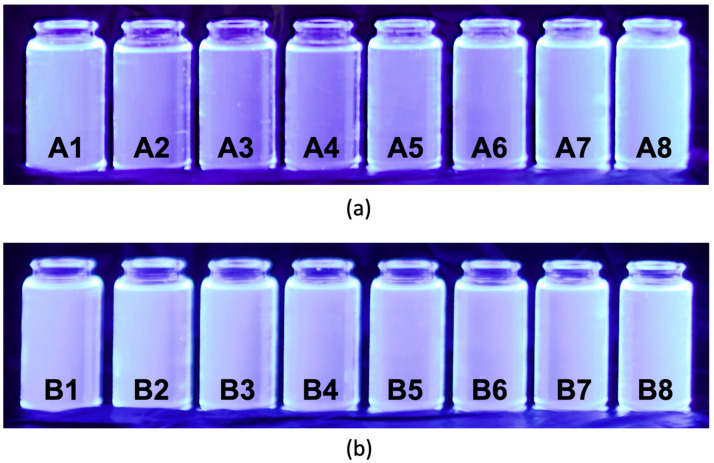
Scintillation light emitted by the LS samples upon UV excitation: (**a**) Eight samples with different PPO concentrations. Left to right: Samples A1 to A8; (**b**) Eight samples with different concentrations of bis-MSB. The PPO concentration is kept constant at 3 g/L. Left to right: Samples B1 to B8.

**Figure 2 sensors-23-02728-f002:**
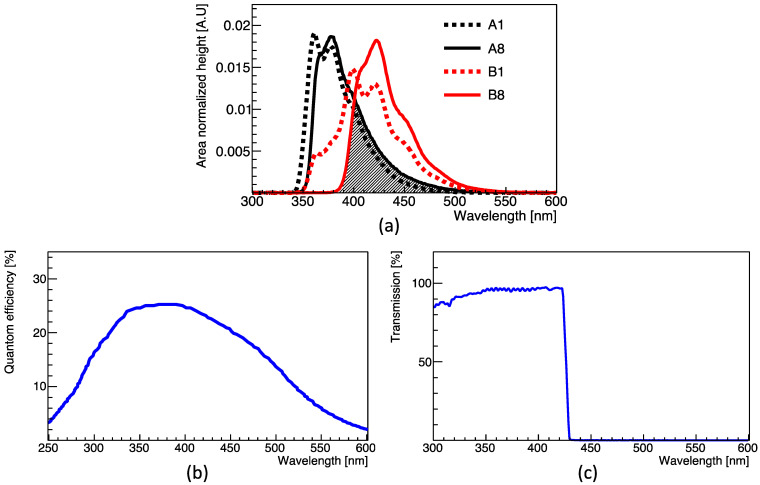
(**a**) Emission spectra of the LS samples with different PPO and bis-MSB concentrations. (Black lines) The PPO concentration is 0.5 g/L (A1) or 10 g/L (A8). (Red lines) The PPO concentration is 3 g/L and the bis-MSB concentration is 1 mg/L (B1) or 50 mg/L (B8); (**b**) Wavelength-dependent photon detection efficiency of the H7195 PMT [17]; (**c**) Transmittance of the short-pass-filter as a function of the wavelength.

**Figure 3 sensors-23-02728-f003:**
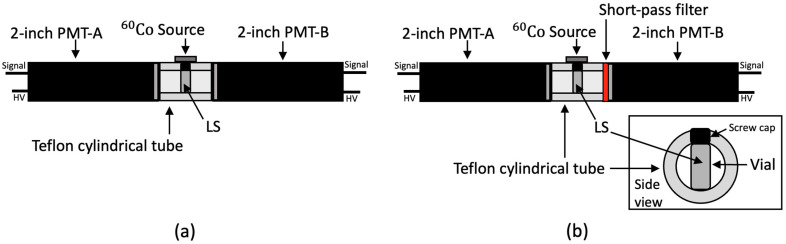
Schematic setup for the scintillation response measurement. The LS contained in the vial emits scintillation light via Compton scattering of the gamma-ray emitted by ^60^Co: (**a**) After fixing a vial containing an LS into the PTFE coupler between the two 2-inch PMTs, the response depending on the changes in the PPO and bis-MSB concentration was measured; (**b**) Setup for measuring the phenomenon that the emission area is cut by the filter using a 425 nm short-pass filter. A short-pass filter was added to the 2-inch PMT-B only.

**Figure 4 sensors-23-02728-f004:**
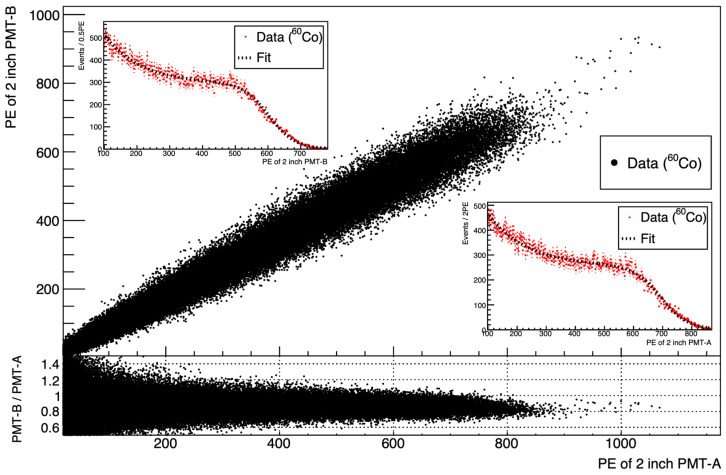
Observed light yield of sample B3, obtained from two H7195 PMTs using the setup shown in Figure 3a. A clear PE correlation is observed between the two PMTs. The inset presents a 1D projection plot for the observed PE. The upper (lower) inset corresponds to the PE distribution of PMT-B (PMT-A). Each PE distribution is fitted to determine the Compton edge.

**Figure 5 sensors-23-02728-f005:**
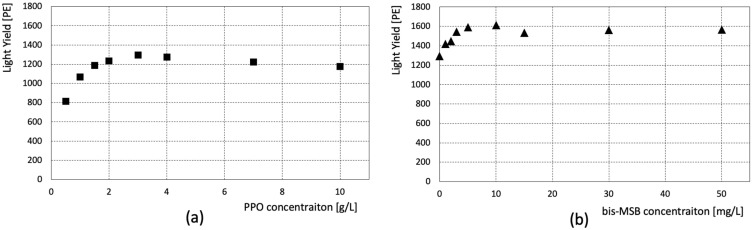
(**a**) Scintillation light yield with different PPO concentrations; (**b**) Light yield with different bis-MSB concentrations. The PPO concentration is constant at 3 g/L in (**b**).

**Figure 6 sensors-23-02728-f006:**
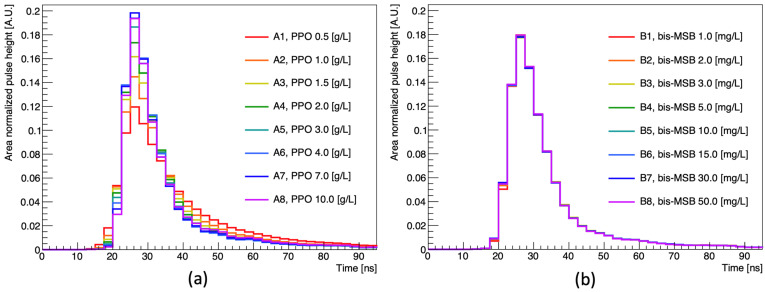
Area normalized waveform with different PPO and bis-MSB concentrations: (**a**) Pulse shape with varying PPO concentration; (**b**) Pulse shape with varying bis-MSB concentration. The PPO concentration is kept constant at 3 g/L. The waveform was obtained by averaging the area normalized pulse shape of 10,000 events.

**Figure 7 sensors-23-02728-f007:**
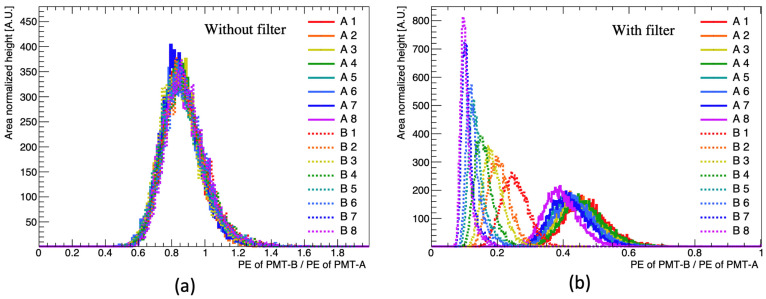
Observed PE ratio between PMT-A and PMT-B at different fluor concentrations: (**a**) Measurement without using the filter. The difference in the photon detection efficiency is estimated by comparing the Compton edges observed by PMT-A and PMT-B in Figure 4; the difference is approximately 15%. The observed PE ratio distribution between PMT-A and PMT-B is shifted to the left; (**b**) Measurement with the filter. The PE ratio distribution varies with the fluor concentration when the filter is used.

**Figure 8 sensors-23-02728-f008:**
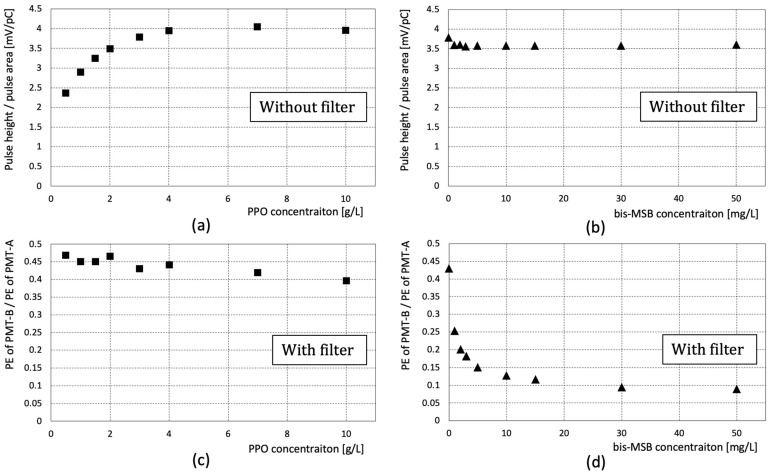
(**a**) Area normalized pulse height with different PPO concentrations; (**b**) Area normalized pulse height with different bis-MSB concentration; The pulse height of Figure (**a**,**b**) were obtained without the short-pass filter; (**c**) The ratio of PE passing through the short-pass filter with different PPO concentrations; (**d**) The ratio of PE passing through the short-pass filter with different bis-MSB concentration. In Figure (**b**,**d**), the PPO concentration is kept constant at 3 g/L. Note that the amount of bis-MSB is less than that of PPO by a factor of ~1000.

**Table 1 sensors-23-02728-t001:** LS samples with different PPO concentrations. The concentration of PPO varied from 0.5 g/L to 10 g/L. The bis-MSB was not dissolved in the samples.

Sample	PPO Concentration (g/L)
A1	0.5
A2	1
A3	1.5
A4	2
A5	3
A6	4
A7	7
A8	10

**Table 2 sensors-23-02728-t002:** LS with different PPO and bis-MSB concentrations. The concentration of PPO was optimized and kept constant at 3 g/L and the concentration of bis-MSB varied from 1 mg/L to 50 mg/L.

Sample	PPO Concentration (g/L)	bis-MSB Concentration (mg/L)
B1	3	1
B2	3	2
B3	3	3
B4	3	5
B5	3	10
B6	3	15
B7	3	30
B8	3	50

## Data Availability

Not applicable.

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
