# Peer review of "Feasibility Study on the Discrimination of Fluor Concentration in the Liquid Scintillator Using PMT Waveform and Short-Pass Filter"

_sensors, 2023, doi:10.3390/s23052728_

Round 1

Reviewer 1 Report

1)line 28 "among the particles predicted by the Standard Model neutrinos.."

This statement is wrong: neutrinos are not predicted by the Standard Model!
Neutrinos was predicted in 1930 by Pauli and detected in 1956.
The standard model was built when neutrino had already been detected and it was
constructed assuming the  massless neutrinos.

2) paragraph starting at line 44
No information is provided on the stability of the scintillator used in this paper.
In references 5 and 6 there are  scintillators loaded with gadolinium different from the one studied in this paper.
The referee believes that more information on the stability should be provided

3)In  Figure 1 as mentioned in the text there is very small difference between the samples.
Authors should consider removing this figure or at least cleaning up  the figure by removing
the area outside the vials

4) line 141  "filter was added to the only 2-inch PMT side"
it's not clear perhaps "filter was added    to the 2-inch PMT-B only"?

5) line 273, why the fluor concentration should change over time and on which time scale?

6) The reference Abusleme et al NIM vol 988 "Optimization of the JUNO liquid
scintillator composition using a Daya Bay" should be included in the references, because
the scintillator is identical to the one studied in this paper

Author Response

Please take a look at an attached [Q/A] file. Thanks. 

Reviewer 2 Report

Please provide additional details on the motivation and required sensitivity of this method to be usable for the described applications.

Conclusions should take into account obtained results and describe the potential of this method taking into account all possible info that can be extracted from waveforms, spectra, light yield.

20: was -> were

36: remove either yield or production

56: of were - > are

64: and is environmentally friendly

70: emitted lights converted -> emitted light is converted

97: Cannon -> Canon

98: wavelength difference -> wavelength difference or better intensity differences?

102: remove “the at”

130: interval -> regime 

131: sensitive -> high

Info given in table 3 is already (almost) fully provided in text. Better to remove table and add all info directly when describing short pass in text.

Paragraph starting 145:

Why was Teflon tube used instead of placing PMTs closer to the LS?

What is the geometric efficiency for collecting light emitted in the LS by the PMT?

Before using the two PMTs with one behind a short pass filter, did you compare the response and can you comment on the agreement between the PMTs?

Instead of giving number of photons detector by each PMT, it is better to normalise light yield by geometric acceptance and QE of PMTs to have a setup independent number. Please either normalise by these numbers or provide values for geometric acceptance and wavelength dependent QE to allow normalisation.

Figure 6: add concentrations next to sample labels

Are waveforms shown in figure 6 single events or averaged waveforms? 

Please describe common concentrations of fluors in LS that should be monitored. It would be good to describe this better in the introduction to quantify the sensitivity that would be required for LS light yield monitoring.

Author Response

(The authors gave the same response as above.)

Round 2

Reviewer 1 Report

The manuscript has been sufficiently improved to warrant publication in Sensors

Reviewer 2 Report

Thank you for addressing questions and comments and for submitted this improved version of the manuscript.